# Urinary Peptides as Potential Non-Invasive Biomarkers for Lupus Nephritis: Results of the Peptidu-LUP Study

**DOI:** 10.3390/jcm10081690

**Published:** 2021-04-14

**Authors:** Maxence Tailliar, Joost P. Schanstra, Tim Dierckx, Benjamin Breuil, Guillaume Hanouna, Nicolas Charles, Jean-Loup Bascands, Bertrand Dussol, Alain Vazi, Laurent Chiche, Justyna Siwy, Stanislas Faguer, Laurent Daniel, Eric Daugas, Noémie Jourde-Chiche

**Affiliations:** 1AP-HM, Centre de Néphrologie et Transplantation Rénale, Hôpital de la Conception, 13005 Marseille, France; maxence.tailliar@ap-hm.fr (M.T.); bertrand.dussol@ap-hm.fr (B.D.); 2Institut National de la Santé et de la Recherche Médicale (INSERM), U1297, Institut of Metabolic and Cardiovascular Disease (I2MC), 31432 Toulouse, France; joost-peter.schanstra@inserm.fr (J.P.S.); benjamin.breuil@inserm.fr (B.B.); faguer.s@chu-toulouse.fr (S.F.); 3Université Toulouse III Paul-Sabatier, 31062 Toulouse, France; 4Laboratory of Clinical and Epidemiological Virology, Department of Microbiology and Immunology, Rega Institute for Medical Research, KU Leuven, 3000 Leuven, Belgium; tim.dierckx@kuleuven.be; 5AP-HP, Service de Néphrologie, Hôpital Bichat, DMU VICTOIRE, 75018 Paris, France; guillaume.hanouna@aphp.fr (G.H.); eric.daugas@aphp.fr (E.D.); 6Centre de Recherche sur l’Inflammation, Université de Paris, INSERM UMRS1149, CNRS ERL8252, Labex INFLAMEX, DHU FIRE, 75890 Paris, France; nicolas.charles@inserm.fr; 7Institut National de la Santé et de la Recherche Médicale (INSERM), U1188-Université de La Réunion, 97490 Saint-Denis, France; jean-loup.bascands@inserm.fr; 8Centre d’Investigation Clinique, CHU Conception, AP-HM, 13005 Marseille, France; alain.vazi@ap-hm.fr; 9Médecine Interne, Hôpital Européen, 13003 Marseille, France; l.chiche@hopital-europeen.fr; 10Mosaiques Diagnostics GmbH, 30659 Hannover, Germany; siwy@mosaiques.de; 11CHU de Toulouse, Service de Néphrologie, 31300 Toulouse, France; 12AP-HM, Laboratoire d’Ananatomie Pathologique, Hôpital de la Timone, 13005 Marseille, France; laurent.daniel@ap-hm.fr; 13Center for CardioVascular and Nutrition Research (C2VN), Aix-Marseille University, INSERM, INRAE, 13005 Marseille, France

**Keywords:** lupus nephritis, systemic lupus erythematosus, non-invasive, biomarker, proteomics, peptidomics, urine, kidney biopsy, classification, prognosis

## Abstract

*Background*: Lupus nephritis (LN) is a severe manifestation of Systemic Lupus Erythematosus (SLE). The therapeutic strategy relies on kidney biopsy (KB) results. We tested whether urinary peptidome analysis could non-invasively differentiate active from non-active LN. *Design*: Urinary samples were collected from 93 patients (55 with active LN and 38 with non-active LN), forming a discovery (*n* = 42) and an independent validation (*n* = 51) cohort. Clinical characteristics were collected at inclusion and prospectively for 24 months. The urinary peptidome was analyzed by capillary-electrophoresis coupled to mass-spectrometry, comparing active LN to non-active LN, and assessing chronic lesions and response to therapy. The value of previously validated prognostic (CKD273) and differential diagnostic (LN172) signatures was evaluated. *Results:* Urinary peptides could not discriminate between active and non-active LN or predict early response to therapy. Tubulo-interstitial fibrosis was correlated to the CKD273. The LN172 score identified 92.5% of samples as LN. Few patients developed new-onset CKD. *Conclusions:* We validated the CKD273 and LN172 classifiers but did not identify a robust signature that could predict active LN and replace KB. The value of urinary peptidome to predict long-term CKD, or renal flares in SLE, remains to be evaluated.

## 1. Introduction

Systemic Lupus Erythematosus (SLE) is a systemic auto-immune disease which can affect many organs. Lupus Nephritis (LN), which occurs in 20–70% of patients according to ethnicity [1], worsens the prognosis of SLE [2] and leads to End Stage Kidney Disease (ESKD) in 10% of patients after 10 years [3]. The treatment of LN is based on histopathological classification [4,5] which mainly distinguishes patients with active proliferative lesions (class III-A or class IV-A ± C ± V LN according to the ISN/RPS classification [5]) from patients with non-active LN (other LN classes). Patients with non-active LN will receive antimalarials and renin angiotensin aldosterone system (RAAS) blockers. Active proliferative LN requires additional immunosuppressive therapy, until remission is achieved, followed by maintenance immunosuppressive therapy to avoid relapses [6,7]. Lesions of tubulointerstitial nephritis (TIN) can be associated to the specific glomerular rearrangements and their impact on LN outcome has been highlighted recently [8]. Interstitial inflammation on initial KB in patients with LN can predict long-term renal survival [9], even among patients with class IV LN. The severity of interstitial fibrosis and tubular atrophy (IF/TA) also independently predicts long-term renal survival. In a murine model of LN (NZB/W mice), the infiltration of renal tubulointerstitium by anti-dsDNA-secreting plasma cells could suggest a specific role of TIN in the onset of LN [10]. Because proteinuria, urinary sediment, serum creatinine or immunological markers (anti-dsDNA, complement fractions) are poorly correlated with pathological lesions and because there is to date no specific biomarker of TIN lesions, a kidney biopsy (KB) is required for every suspected flare of LN [11]. Nevertheless, KB is associated with significant complications [12] such as macroscopic hematuria, pain, hematoma, or in exceptional cases, nephrectomy or death. As such, there is a need for non-invasive biomarkers, such as urinary biomarkers, to predict the pathological severity of LN and to predict the response to therapy and the risk of chronic kidney disease (CKD) in patients with LN.

For example, promising predictive values for LN activity/severity and/or the prediction of renal flares have been reported for urinary cytokines or markers of tubular injury, either alone [13] or combined [14], or the quantification of inflammatory cell markers [15]. However, the validation of these urinary biomarkers in independent cohorts is still awaited. Because LN pathogenicity is a complex problem, it is likely that a combination of non-invasive biomarkers is to be preferred [16]. As such, urinary proteomics or peptidomics (i.e., small molecular weight proteins) is an appealing approach for the evaluation of local tissue inflammation and damage.

The analysis of urinary peptides shows a strong reproducibility under different storage conditions [17]. Using capillary-electrophoresis coupled to mass-spectrometry (CE-MS), urinary peptidome analysis has been used as a prognostic or diagnostic tool in fetal medicine [18,19,20], in CKD with the use of the urinary peptide CKD273 classifier [20,21,22,23] and in different glomerular diseases [24,25,26]. In the field of LN, previous studies investigating the urinary proteome used older technologies, small study populations or lacked validation cohorts [27,28,29,30,31,32]. Studies focusing on the urinary peptidome did not consider the difference between active LN and non-active LN, and instead aimed to discriminate LN from healthy controls or from other renal diseases [33,34]. The LN172 urinary peptide classifier, in particular, was able to differentiate LN from CKD with other etiologies (focal segmental glomerulosclerosis, minimal-change disease, membranous nephropathy, diabetic nephropathy, hypertensive nephrosclerosis and vasculitis-induced kidney disease) with a good accuracy (AUC 0.82) [35].

In the present study, we aimed to assess, using urinary peptidomics, whether we could identify urinary peptides that discriminated between active and non-active LN, in patients undergoing a KB for a suspicion of LN flare. The secondary objectives were to investigate if the urinary peptidome could predict: (1) glomerular chronic lesions, (2) IF/TA, (3) renal response to treatment, (4) renal flares and (5) renal function after 24 months. We also evaluated the performances of the previously established CKD273 and LN172 classifiers in this cohort.

## 2. Materials and Methods

### 2.1. Patients’ Characteristics

Patients were included from July 2011 to December 2016 in 3 French centers participating in the “Groupe Coopératif sur le Lupus Rénal” (GCLR). The patients of the discovery cohort (*n* = 42) were included from July 2011 to November 2012, and the patients of the validation cohort (*n* = 51) were included from September 2014 to December 2016. All patients signed an informed consent before any study procedure, and the samples were included in the biological collections of these 3 centers (Marseille: DC 2012-1704; Paris Bichat: ID-RCB 2014-A00809-38; and Toulouse: DC 2011-1388). Inclusion criteria were: age ≥ 18 years, SLE according to the ACR 1997 [36] or SLICC 2012 [37] classification criteria, urinary sample collected at the time of KB, biopsy including ≥10 glomeruli. Clinical data and laboratory results were collected on the day of KB, and during the follow-up.

Initial clinical characteristics included age, sex and ethnicity. Serum creatinine and estimated-Glomerular Filtration-Rate (eGFR) using the MDRD formula [38], urinary protein/creatinine ratio (UPCR), urinary sediment, complement and anti-dsDNA antibodies were collected at the time of KB and during the longitudinal follow-up. Therapeutic regimens for SLE before the renal flare, at the time of KB, and for the flare of LN, were recorded. The renal response to treatment was evaluated using eGFR and UPCR at M6, M12 and M18, and the long-term renal function was estimated from the same parameters measured at M24 and at the end of follow-up. Remission was defined with 3 possible thresholds of UPCR (0.2 g/g, 0.5 g/g or 0.7 g/g), together with a return to baseline eGFR or a decrease in eGFR not exceeding 20%. CKD was defined by a persistent decline of eGFR < 60 mL/min/1.73 m^2^. LN relapse was defined as a new KB showing active lesions and leading to therapeutic implementation.

### 2.2. Kidney Biopsy Pathological Analysis

Pathological data were interpreted according to the ISN/RPS 2003 classification [5], with the addition of IF/TA evaluation (F0: ≤5%; F1: 6–25%; F2: 26–50%; F3: >50%) according to the Banff classification [39]. Active LN was defined as class III or IV LN with active lesions, ± associated class V. Non-active LN was defined as class II LN, isolated class V LN or the presence of chronic lesions only.

### 2.3. Sample Preparation

Fresh urine was collected on the morning of the kidney biopsy, centrifuged to remove the cell pellet, and stored at −80 °C. Urine samples were thawed immediately before use. A volume of 0.7 mL was diluted with 0.7 mL 2 M urea, 10 mM NH4OH and 0.02% sodium dodecyl sulphate (SDS). In order to remove high molecular weight polypeptides, samples were filtered using Centrisart ultracentrifugation filter devices (20 kDa molecular weight cut-off); Sartorius, Goettingen, Germany) at 3000× *g* until 1.1 mL of filtrate was obtained. The filtrate was desalted with PD-10 column (GE Healthcare, Sweden) equilibrated in 0.01% NH4OH in HPLC-grade water. The prepared samples were lyophilized and stored at 4 °C. Shortly before CE-MS analysis, lyophilized samples were resuspended in HPLC-grade water (Merck KGaA, Darmstadt, Germany). The preparation method has previously been described in more detail [40].

### 2.4. CE-MS Analysis and Data Processing

CE-MS analysis was performed as previously described [41]. Briefly, CE-MS analyses were performed using a Beckman Coulter Proteome Lab PA800 capillary electrophoresis system (Beckman Coulter, Fullerton, CA, USA) on-line coupled to a micrOTOF II MS (Bruker Daltonic, Bremen, Germany). The electro-ionization sprayer (Agilent Technologies, Palo Alto, CA, USA) was grounded, and the ion spray interface potential was set to −4.5 kV. Data acquisition and MS acquisition methods were automatically controlled by the CE via contact-close-relays. Spectra were accumulated every 3 s, over a range of m/z 350 to 3000. In the next step the MosaiquesVisu software package was applied to deconvolute mass spectral ion peaks, because ionization produced ions at different charged states from the original urinary peptides. MosaiquesVisu was used to deconvolute mass spectral peaks representing identical molecules into singles masses. The obtained peak list of each polypeptide is characterized by molecular mass (in Daltons), CE-migration time (in minutes), and normalized ion signal intensity. MS signal intensities were used as measure of relative abundance and normalized using 29 internal standard peptides as described by Jantos-Siwy et al. [42]. All detected peptides were deposited, matched and annotated in a Microsoft SQL database, permitting further correlation and statistical analysis. Raw peptide data can be found in Appendix A.

### 2.5. Biomarker Selection and Modelling

For the identification of new candidate urinary biomarkers for LN activity, the reported *p*-values were calculated using the Wilcoxon rank sum test (R software, version 3.1.3) comparing active LN to non-active LN samples followed by adjustment for multiple testing (Benjamini and Hochberg, 1995). Peptides that were detectable in at least 60% of the samples in one of the two groups (active LN or non-active LN) and reached an adjusted *p-*value of <0.05 were considered as potential biomarker peptides. The R package randomForest (version 4.6-14) was used to generate biomarker models that consisted of an ensemble of 500 trees sampling 5 predictors per split. The overall yield of the polypeptide pattern was evaluated by receiver operating characteristic (ROC) and area under curve (AUC) plots using the Prism 7.00 GraphPad software.

We also tested previously validated peptidome classifiers, using the same algorithms as in the original publications: CKD273 [21,22,43] as a continuous variable, and LN172 both as a continuous and categorical variable [35].

## 3. Results

### 3.1. Patients’ Characteristics

The whole cohort comprised 86 women (92.5%) and 7 men (7.5%). Median age was 35 years, most patients were of European (43%) or North African (24.7%) origin. The demographic and renal characteristics at baseline are described in Table 1. There was a majority of active LN (59.1% of class III or IV ± V with active lesions). A large majority of patients had previously received hydroxychloroquine (HCQ) (80.6%) or corticosteroids (CS) (77.4%), and 33.3% had been treated with Cyclophosphamide. At the time of KB, 68.8% of patients were prescribed HCQ, 70.1% CS, 17.2% mycophenolate mofetil (MMF) and 9.7% Azathioprine (AZA).

The mean follow-up was 43.2 months; 9 patients were lost to follow-up (Table 2). At least one relapse of LN occurred in 19 (20.4%) patients after a median of 12 months; 68.4% of relapses occurred within two years of the previous flare. At the end of the follow-up, 12 (12.9%) patients displayed CKD, among whom 9 had preexisting CKD at inclusion (only 3 patients developed new-onset CKD); 5 patients progressed to ESKD requiring renal replacement therapy (4 renal transplantations, 1 hemodialysis); 4 (4.3%) patients died during the follow-up.

There was no significant difference in gender, age, ethnicity or renal function between the discovery cohort and the validation cohort (Appendix B, Table A1), except for a difference in the center of origin (more patients from Paris Bichat in the discovery cohort, more patients from Marseille in the validation cohort).

### 3.2. Absence of Urinary Peptides Predicting Proliferative LN

In the discovery cohort of 42 samples, comprising 22 samples from patients with active LN and 20 samples from patients with non-active LN, we identified 88 significant peptides (*p* < 0.05, Wilcoxon) that differentiated active from non-active LN. However, these peptides lost significance after multiple testing correction suggesting limited discriminative power of these peptides. This was confirmed by attempts to validate these peptides in the validation cohort (comprising 33 patients with active LN and 18 with non-active LN). While a random forest model of these 88 peptides performed well in the discovery cohort (area under the curve (AUC) of 0.834), it performed poorly in the validation cohort (AUC 0.542). The list of these 88 peptides is provided in Appendix A.

Assuming that glomerular activity was a continuous dimension, we hypothesized that the limit of our analysis may be the dichotomous distinction between active and non-active LN and that urinary peptidome may be correlated to the amount of active glomerular lesions, but Spearman correlation analysis showed no peptide was significantly correlated to glomerular activity after multiple testing correction.

Routine laboratory parameters were also tested as potential discriminative markers between active LN and non-active LN (Table 3). Complement consumption, anti-dsDNA antibodies and pyuria were significantly associated to active LN (*p* = 0.001, *p* = 0.007 and *p* = 0.03, respectively). In non-active LN, serum creatinine was significantly lower (87.1 ± 71 versus 96.7 ± 71; *p* = 0.01) and eGFR significantly higher (106.7 ± 45.8 versus 85.9 ± 31; *p* = 0.009). Anti-dsDNA antibodies offered the best sensitivity (96%) and pyuria the best specificity (67%) for active LN. UPCR and hematuria did not differ significantly between active and non-active LN.

### 3.3. The Urinary Peptidome Does Not Predict Glomerulosclerosis

Because the urine peptidome can reflect glomerular rearrangements and fibrosis [44], we hypothesized that severe glomerular sclerosis may significantly change the composition of peptides found in patients developing LN. We tried to identify a peptide signature able to predict glomerular chronicity defined by glomerulosclerosis ≥ 25% (or ≥ 50% in a second analysis). After the multiple testing correction, 177 peptides were predictive of glomerular chronicity in the discovery cohort, but none of them could be confirmed in the validation cohort. The list of these 177 peptides is provided in Appendix A.

### 3.4. CKD273 Is Correlated to Tubulo-Interstitial Chronicity

No peptide profile appeared reliable to predict tubulo-interstitial chronicity or its severity, but a significant correlation was found between the level of IF/TA and the CDK273 score (r = 0.3139; *p* = 0.0015) (Figure 1). Several clinical characteristics were strongly correlated to IF/TA (Table 4), such as SLE duration (r = 0.408; *p* < 0.0001) and LN duration (r = 0.450; *p* < 0.0001). Age was also associated to IF/TA (r = 0.231; *p* = 0.026). No correlation was found with hematuria or UPCR, but the level of IF/TA at inclusion was correlated with serum creatinine and eGFR (r = 0.274; *p* = 0.008 and r = −0.290; *p* = 0.0049, respectively). The application of CKD273 with the established cut-off for early CKD detection [45,46] scored 82 LN patients of the whole cohort (88.2%) as a high risk group for CKD development. The eleven patients that scored negative (classification score below 0.154) had a higher mean eGFR of 112.5 (SD = 40.2) than the high risk group patients (mean eGFR = 88.6 (SD = 38.4), *p* = 0.057).

### 3.5. LN 172 Is a Sensible Predictor for LN

The previously established LN172 classifier [35] identified 86 out of the 93 patients as patients with LN (LN172+, sensitivity 92.5%). The diagnostics threshold of 0.013 was established using the independent validation set (*n* = 474) of Siwy et al. data [35] and resulted there in a sensitivity of 79.3% and specificity of 75.1% (AUC = 0.82). Among the 7 LN patients who were not identified as LN patients by the LN172 classifier (LN172-), 6 had active LN and 1 non-active LN. The LN172- patients had lower eGFR at inclusion (46.2 ± 29 vs. 94.9 ± 38 mL/min/1.73 m^2^; *p* = 0.001) and at the end of the follow-up (67.5 ± 32 vs. 104.8 ± 30 mL/min/1.73 m^2^; *p* = 0.004), and were more likely to display baseline CKD (42.9% vs. 7%) than LN172+ patients. Their CKD273 score was significantly higher (*p* = 0.02) (Appendix B, Table A2). Accordingly, the LN172 score was inversely correlated with the CKD273 score in the global cohort (Pearson correlation, r = −0.3; *p* = 0.004).

### 3.6. The Urinary Peptidome Does Not Predict Early Remission

We hypothesized that urinary peptides may predict early renal remission and thus renal response to treatment. No peptide profile was significantly predictive of renal remission at M6, whatever the UPCR threshold used to define remission (0.2, 0.5 or 0.7 g/g) (Figure 2).

The relation between urinary peptidome and long term renal evolution could not be tested as only three patients developed new-onset CKD during follow-up. As such, statistical analysis could not be performed to predict renal long-term evolution. 

## 4. Discussion

In this study, comprising an independent discovery and a validation cohort, no urinary peptide signature could discriminate active LN from non-active LN. Previous works aimed to discriminate LN from other causes of renal damage [35] or renal SLE from non-renal SLE [34]. To our knowledge, this is the first study using urinary peptidomics to address the specific distinction between active and non-active LN, which is crucial for the therapeutic strategy in patients with LN.

Other studies have identified significant changes in urinary proteome in the course of LN. For instance, Zhan et al. [28] highlighted the modification of the urinary low-molecular weight proteins during a flare cycle (baseline, preflare, flare, post-flare) in sequential analyses by SELDI-TOF of urinary samples from 19 patients. Though these results have not been validated in an independent cohort, this work supported the intuitive hypothesis that molecular changes occur during the different phases of LN and that urine proteome/peptidome analysis could allow the diagnosis of severe flares. Wei et al. identified 300 peptides discriminating LN from non-renal SLE and controls [34]. The LN172 urinary peptide classifier developed by Siwy et al. [35] aimed to discriminate LN from other kidney diseases. In a cohort comprising 92 patients with LN and 1088 patients with other kidney diseases, the LN172 score correctly classified LN samples with an AUC of 0.82 and sensitivity of 79.3% and specificity of 75.1%. We confirmed in an independent cohort that, indeed, the LN172 classifier displayed a high sensitivity to detect LN (92.5%). Interestingly, the 7 patients that were not classified as LN by the LN172 classifier had lower eGFR and 3 of the 7 patients displayed CKD at baseline (eGFR < 60 mL/min/1.73 m^2^). In the setting of more advanced kidney disease, one can hypothesize that the urinary peptidome from these 7 patients was enriched with “regular CKD” peptides and thereby modifying its LN specific profile. This is confirmed by the fact that the LN172- patients had a higher CDK273 score, which targets chronic kidney rearrangements, and the inverse correlation between these two classifiers reinforces this hypothesis.

A possible pitfall for urinary peptidome analysis in this work, as in real life, is the baseline immunosuppressive therapy of some patients at the time of KB that could mitigate some markers of active LN. Because LN is not always diagnosed early, and 45% of patients from this study had a previous history of LN flare, our second hypothesis was that urinary peptidome analysis would reflect the severity of chronic kidney damage, either glomerular or tubulo-interstitial. Indeed, the urinary peptidome has been tested and validated through the development of the CDK273 classifier in CKD diagnosis and prediction [22], and in diabetic kidney disease (early prediction, diagnosis and CKD prediction) [47]. However, in this work, no robust “SLE-specific” peptide signature predicted glomerular chronicity or tubulo-interstitial fibrosis. The application of CKD273 for the early detection of high-risk patients like performed by Tofte et al. resulted in 88.2% high risk patients for CKD development in our cohort. Moreover, CKD273 signature was positively correlated to the level of IF/TA in patients with LN, supporting the hypothesis that peptide markers can reflect chronic TIN rearrangement. While the therapeutic guidelines focus on glomerular lesions through the ISN/RPS classification, TIN lesions, whether acute (such as tubulitis) or chronic (such as IF/TA), are strong independent prognostic factors of kidney survival in LN [9,48,49]. In particular, in a study from Hsieh et al., interstitial nephritis was found to be a better predictor of renal failure than the ISN/RPS class [50]. The association between the severity of TIN lesions and the ISN/RPS class is inconstant [50,51] which suggests that they may result from independent inflammatory pathways and thus may require specific therapeutic adjustments. Taken together, these results explain the better prognostic performance of global histologic assessments compared to the glomerulo-centric ISN/RPS classification [51,52]. The association between the CKD273 score and IF/TA encourages the development of the urinary peptidome analysis as a prognostic factor through the non-invasive assessment of TIN lesions. In our study, despite a minimum follow-up of 2 years (with a median of 4 years), we could not test the predictive value of urinary peptidome for the development of CKD, since only 3 patients developed new-onset CKD after a mean follow-up of 4 years. Moreover, it can be noted that SLE vintage and LN duration, as well as serum creatinine were better predictors of renal fibrosis than CKD273 in this cohort.

Another possible pitfall that could blunt the predictive value of urinary peptidome analysis in LN is the abundance of proteinuria. Mean UPCR was over 2 g/g in both the discovery and validation cohorts in the present study. It is possible that high proteinuria is a confounding factor on the relation between the peptidome and the clinical classification of the patients. However, among the 177 peptides which appeared correlated to glomerular chronicity in the discovery cohort, only 6 were correlated to UPCR. Similarly, among the 240 peptides identified by Wei et al. [34] as significantly increased in LN compared with non-renal SLE patients and controls, only 13 were correlated to proteinuria. In addition to proteinuria, the low number of patients used in the discovery set (*n* = 42) may have limited the chance to find robust discriminatory peptides between the active and non-active form of LN.

Anti-dsDNA antibodies, complement consumption and pyuria were significantly associated to active LN, but with poor diagnostic performances. A combination of 12 clinical and biological markers has been previously used to predict active LN [53], with a ROC curve AUC of 0.83, offering a fine orientation but still insufficient to replace KB.

The poor correlation of molecular markers and renal pathological results described here has also been reported within the renal tissue, in transcriptomic analysis of dissected glomeruli and tubules from KB of patients with LN [54,55]. Single-cell RNA sequencing from KB of patients with LN and controls [56] has shown specific gene expression profiles associated with LN, strongly correlated to the profile found in urinary cells. Similarly, Abedini et al. recently demonstrated that the single-cell transcriptome analysis could detect and quantify almost all kidney cells in urinary samples from patients with diabetic kidney disease (DKD) and controls and that the distribution of the detected cells was different between DKD and controls [57]. These results suggest a possible use of urinary cells transcriptomics to identify immunological pathways activated within the kidney and assess non-invasively infra-clinical rearrangements occurring in the course of CKD.

Although urinary peptidomics did not differentiate between active LN and non-active LN patients in this cohort, there are other potential applications of urinary peptidomics in LN that were not explored in this study. First, early detection of LN flare through the detection of early renal rearrangements could be investigated using urinary peptidomics. Some immunological parameters are associated with the risk of LN in SLE [58,59,60] but remain poorly predictive of flares [61,62]. Moreover, some patients show active renal lesions while no proteinuria, hematuria or pyuria is detected [63,64]. Here, all patients were sampled while clinically flaring and had a renal biopsy because of proteinuria. Second, as insufficient data on CKD and ESKD were available, their relation to the value of urinary peptidomics also remains untested.

## 5. Conclusions

We were not able to identify a urinary peptide profile predictive of active LN, of glomerular activity or chronicity in this cohort of patients with LN. However, the LN172 signature displayed high sensitivity for the diagnosis of LN while the CKD273 classifier was correlated to IF/TA which paves the way to the development of a new prognostic tool in the care of patient with LN. Whether urinary peptidomics will allow the early detection of renal flares in SLE remains to be determined. To date, kidney biopsy remains the cornerstone of LN management and could provide useful information on the molecular pathways activated at a given time, in addition to the pathological lesions observed [7]. Nonetheless, non-invasive biomarkers such as the urinary peptidome are still forecasted to prove their added-value for a more personalized care of patients with LN, yet their development and validation will require larger cohorts.

## Figures and Tables

**Figure 1 jcm-10-01690-f001:**
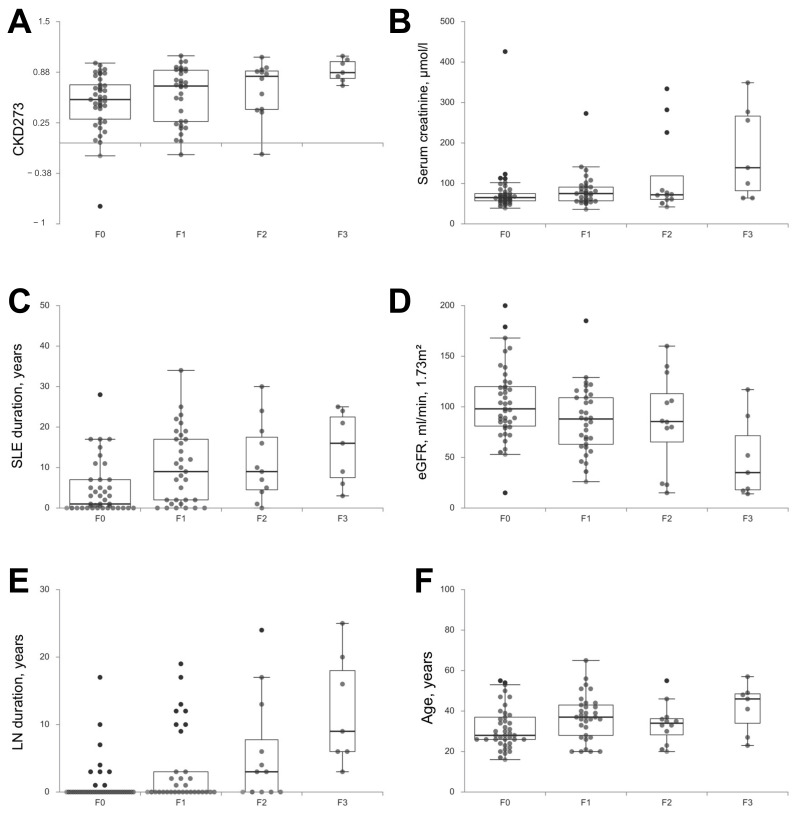
Correlation between Interstitial Fibrosis/Tubular Atrophy and (**A**) CKD273, (**B**) serum creatinine at inclusion, (**C**) SLE duration, (**D**) eGFR at inclusion, (**E**) LN duration and (**F**) age. eGFR: estimate glomelural filtration rate calcutated with the MDRD formula; LN: Lupus Nephritis; SLE: Systemic Lupus Erythematosus.

**Figure 2 jcm-10-01690-f002:**
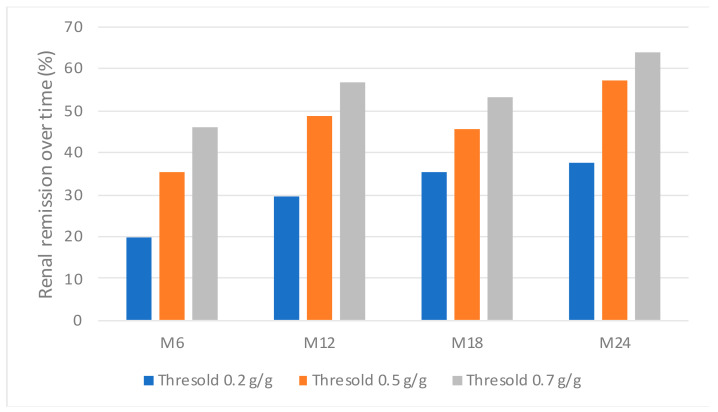
Renal remission over time according to three different thresholds for proteinuria.

**Table 1 jcm-10-01690-t001:** Patients’ baseline characteristics.

Characteristics	
Patients, *n*	93
Women—*n* (%)	86 (92.4)
Age—years	35 ± 11
Range—years	19–58
Ethnicity—*n* (%)	
European	40 (43)
North African	23 (24.7)
African	23 (23.7)
Asian	8 (8.6)
Characteristics at kidney biopsy	
Serum creatinine—μmol/L	92.1 ± 68.5
eGFR—mL/min/1.73 m^2^	91.4 ± 39.2
UPCR—g/g	2.34 ± 2.23
SLE vintage	
First flare of SLE—*n* (%)	21 (22.6)
Pre-existing SLE—*n*%	72 (77.4)
Disease duration—years	10.6 ± 8.5
First renal flare—*n* (%)	55 (59.1)
LN duration—years	8.2 ± 6.9
Kidney biopsies—*n* (%)	93
Class ISN/RPS 2003	
I	1 (1.1)
II	6 (6.5)
III-A or -A/C ± V	22 (23.7)
III-C ±V	5 (5.4)
IV-A or -A/C ± V	33 (35.5)
IV-C ±V	2 (2.2)
Pure V	21 (22.6)
VI	3 (3.2)
Group	
Active LN	55 (59.1)
Non-active LN	38 (40.9)
IF/TA	
F0	41 (44.1)
F1	33 (35.5)
F2	12 (12.9)
F3	7 (7.5)
Previous treatment for SLE (%)	
Hydroxychloroquine	80.6
Corticosteroids	77.4
MMF/Mycophenolic acid	25.8
Azathioprine	19.4
Cyclophosphamide	33.3
Rituximab	8.6
Other	14
Treatment for SLE at inclusion (%)	
Hydroxychloroquine	68.8
Corticosteroids	70.1
MMF/Mycophenolic acid	17.2
Azathioprine	9.7
Cyclophosphamide	1.1
Rituximab	1.1
Other	2.2
Patients/center—*n* (%)	
Marseille	37 (39.8)
Paris	53 (57)
Toulouse	3 (3.2)

Continuous values expressed in mean value ± SD. MMF, mycophenolate mofetil; eGFR, estimated glomerular filtration rate; UPCR, urine protein to creatinine ratio; SLE, Systemic Lupus Erythematosus; LN, Lupus nephritis.

**Table 2 jcm-10-01690-t002:** Outcomes of patients from the discovery and validation cohorts.

	Discovery Cohort	Validation Cohort	*p*
Follow-up—months, median [IQR]	63.5 (51–69)	32 (24–39)	<0.001
Renal function at M24			
Serum creatinine—μmol/L, mean ± SD	64.8 ± 12.5	75.3 ± 19.9	0.12
eGFR—mL/min/1.73 m^2^, mean ± SD	109 ± 22.1	96.6 ± 23.8	0.12
UPCR—g/g, mean ± SD	0.55 ± 0.57	0.42 ± 0.36	0.73
Renal function at last follow-up			
Serum creatinine—μmol/L, mean ± SD	73.5 ± 23.1	76.2 ± 20.8	0.31
eGFR—mL/min/1.73 m^2^, mean ± SD	106.8 ± 26	96.1 ± 19.6	0.22
UPCR—g/g, mean ± SD	0.56 ± 0.55	0.37 ± 0.35	0.44
Relapse of LN			
Total—*n* (%)	11 (26.2)	8 (15.7)	0.21
Early relapse <M24—*n* (%)	5 (11.9)	8 (15.7)	1
Time until relapse—months, mean ± SD	29.3 ± 19.6	12.3 ± 3.6	0.016
CKD (eGFR < 60 mL/min/1.73 m^2^)			
Total, *n* (%)	6 (14.3)	6 (11.8)	0.56
New onset CKD at last follow-up	2 (4.8)	1 (2)	0.58
New onset CKD at M24	0	0	NS
Time until CKD—months, mean ± SD	27.5 ± 1.5	35	NS
ESKD			
At M24—*n* (%)	2 (4.8)	0	NS
At the end of follow up—*n* (%)	3 (7.1)	2 (3.9)	NS
Time until ESKD—months, mean± SD	17.3 ± 8.4	23 ± 11	0.55
Death—*n* (%)	3 (7)	1 (2)	
Time until death—months mean± SD	26.3 ± 13.1	39	NS

CKD: Chronic Kidney Disease; eGFR: estimated glomerular filtration rate; ESKD: End Stage Kidney Disease; LN: Lupus nephritis; M24: 24 months of follow-up; UPCR: urine protein to creatinine ratio.

**Table 3 jcm-10-01690-t003:** Comparison of biological parameters between patients with active and non-active lupus nephritis and discriminative power of routine SLE markers.

	Active LN	Non-Active LN	*p*	Sensitivity	Specificity	PPV	NPV
Peptidomic	No profile			NA	NA	NA	NA
Complement consumption (%)	84	46.1	0.001	0.84	0.54	0.78	0.64
Anti-DNA antibodies (%)	95.8	71.9	0.007	0.96	0.28	0.67	0.82
Hematuria (%)	53.1	40.0	0.37	0.53	0.60	0.68	0.44
Pyuria (%)	64.3	33.3	0.03	0.64	0.67	0.77	0.52
Mean eGFR (mL/min/1.73 m^2^)	82.7 ± 31	104.4 ± 47	0.01	NA	NA	NA	NA
Mean UPCR (g/g)	2.63 ± 2.3	1.97 ± 1.7	0.09	NA	NA	NA	NA
Mean serum creatinine (μmol/L)	96.7 ± 71	87.1 ± 71	0.01	NA	NA	NA	NA

eGFR: estimated glomerular filtration rate, calculated with the MDRD formula; NA: not applicable; NPV: negative predictive value; PPV: positive predictive value; SLE: Systemic lupus erythematosus; UPCR: Urine Protein to Creatinine Ratio.

**Table 4 jcm-10-01690-t004:** Correlation between peptidomic and clinico-biological characteristics with the severity of IF/TA.

Parameter	*r*	*p*
Peptidomic	No profile	
CKD273	0.314	0.0015
SLE duration	0.408	<0.0001
LN duration	0.450	<0.0001
Age	0.231	0.026
Serum creatinine	0.274	0.008
eGFR (MDRD)	−0.290	0.005
UPCR	0.164	0.121
Hematuria	−0.031	0.783

LN: Lupus nephritis; SLE: Systemic lupus erythematosus; UPCR: Urine Protein to Creatinine Ratio.

## Data Availability

The authors confirm that the data supporting the findings of this study are available within the article and its Appendix A. Complementary data are available on request from the corresponding author, [N.J.C].

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
