# Peer review of "Urinary Peptides as Potential Non-Invasive Biomarkers for Lupus Nephritis: Results of the Peptidu-LUP Study"

_jcm, 2021, doi:10.3390/jcm10081690_

Round 1
Reviewer 1 Report
This article by Tailliar et. al describes the suitability or not of Urinary peptides as potential non-invasive biomarkers for lupus nephritis.
While majority of the data is presented well and of negative prognostic value, it would benefit the viewers if presented in a positive note. While uriniary peptides could not provide a robust signature that could predict active LN and replace renal biopsies, especially in context of glomerular injury, there was a strong correlation between Tubulo-interstitial fibrosis and CKD273.
This is an important observation as in LN the extent of tubule-interstitial injury is a better prognostic indicator of progression to renal failure . Tubulointerstitial inflammation, fibrosis and tubular atrophy strongly correlate with poor renal outcomes independent of the extent of glomerular damage. As rightly alluded in the discussion enhanced glomerular permeability due to glomerular injury can lead to overabsorption of proteins triggering tubulointerstitial inflammation, scarring and renal function deterioration.
I think the authors should bring out or highlight the importance of tubuloicntersitital injury in LN and re-write the manuscript that urinary peptides may provide valuable information on the extent of tubulointerstitial injury and hence ensuing progression to ESRD.
Author Response
Response to Reviewer 1 Comments
Dear reviewer,
Thank you for your precious time in reviewing our paper, and for your encouraging remarks. Your valuable and insightful comments led us to improve the manuscript, and present our results with a more positive tone. Please find below our detailed point-by-point response.
Sincerely,
Dr Maxence TAILLIAR & Pr Noémie JOURDE-CHICHE
Point 1: I think the authors should bring out or highlight the importance of tubulointerstitial injury in LN and re-write the manuscript that urinary peptides may provide valuable information on the extent of tubulointerstitial injury and hence ensuing progression to ESRD.
Response 1: We completely agree, and thank reviewer 1 for this suggestion. We highlighted the importance of the assessment of acute and chronic tubulointerstitial (TIN) lesions as a prognostic marker of lupus nephritis (LN).
In the introduction, we presented the general context of the impact of TIN rearrangements in LN citing the work from Wilson & al. who identified interstitial inflammation and interstitial fibrosis and tubular atrophy (IF/TA) as strong independent prognostic factors of renal survival (Page 2 ; Lines : 56-57 and 59-60).
In the discussion, we developed these arguments focusing on the potential prognostic value of the non-invasive evaluation of TIN lesions by the urinary peptidome analysis. In particular, we cited the work from Hsieh & al. who singled out interstitial nephritis and IF/TA as stronger prognostic factors than traditional acute and chronic glomerular lesion (Page 10 ; Lines : 315-330).
In view of this new discussion, we added an opening for the urinary peptidome analysis as a promising prognostic marker in LN in the conclusion (Page 11 ; Lines : 379-382).
The new references added to the manuscript to highlight these different axes are :
8. Wilson, Parker C et al. “Interstitial inflammation and interstitial fibrosis and tubular atrophy predict renal survival in lupus nephritis.” Clinical kidney journal vol. 11,2 (2018): 207-218. doi:10.1093/ckj/sfx093
46. Rijnink, Emilie C et al. “Clinical and Histopathologic Characteristics Associated with Renal Outcomes in Lupus Nephritis.” Clinical journal of the American Society of Nephrology : CJASN vol. 12,5 (2017): 734-743. doi:10.2215/CJN.10601016
47. Obrișcă, B et al. “Histological predictors of renal outcome in lupus nephritis: the importance of tubulointerstitial lesions and scoring of glomerular lesions.” Lupus vol. 27,9 (2018): 1455-1463. doi:10.1177/0961203318776109
48. Hsieh, Christine et al. “Predicting outcomes of lupus nephritis with tubulointerstitial inflammation and scarring.” Arthritis care & research vol. 63,6 (2011): 865-74. doi:10.1002/acr.20441
49. Leatherwood, Cianna et al. “Clinical characteristics and renal prognosis associated with interstitial fibrosis and tubular atrophy (IFTA) and vascular injury in lupus nephritis biopsies.” Seminars in arthritis and rheumatism vol. 49,3 (2019): 396-404. doi:10.1016/j.semarthrit.2019.06.002
50. Pagni, Fabio et al. “Tubulointerstitial lesions in lupus nephritis: International multicentre study in a large cohort of patients with repeat biopsy.” Nephrology (Carlton, Vic.) vol. 21,1 (2016): 35-45. doi:10.1111/nep.12555
Reviewer 2 Report
Tailliar M, et al. investigated the urinary proteomics of the patients with biopsy-proven lupus nephritis (LN) to discriminate active LN from non-active LN, but no significant proteomic signature has been found. Meanwhile, they validated the clinical usefulness of CKD 273 and LN172, previously reported urinary proteomic classifiers for CKD and LN for the prediction of interstitial fibrosis and for the diagnosis of LN, respectively. The manuscript is well written and organized and the conclusions are clear and appropriate.
To make the manuscript more comprehensive and instructive for readers, reviewer would like to request the authors to revise the manuscript on the following points
<Major points>
- In the results, the authors identified 88 significant peptides that were speculated to discriminate active LN from non-active LN (Results 3.2., p5-6), and 177 peptides that were predictive for glomerular chronicity (Results 3.3, p6). The lists of those peptide classifiers should be demonstrated in tables (or in supplement tables), in which they are arranged by their categories. These information may give some ideas to the readers even though they were not significant statistically.
<Minor points>
- In Figure 1, the font size of each graph is small and difficult to read. Please make it larger.
- In some references (ref, 5, 18, 31, 37, 51), the bibliographic information is missing.
Author Response
Response to Reviewer 2 Comments
Dear reviewer,
Thank you for your precious time in reviewing our paper and providing valuable comments. In particular, we agree that the lists of peptides could be of interest for the readers, and now provide it as supplementary data in the new version of our manuscript. Please find below our point-by-point responses.
Sincerely,
Dr Maxence TAILLIAR, Dr Joost SCHANSTRA & Pr Noémie JOURDE-CHICHE
Comments
<Major points>
- In the results, the authors identified 88 significant peptides that were speculated to discriminate active LN from non-active LN (Results 3.2., p5-6), and 177 peptides that were predictive for glomerular chronicity (Results 3.3, p6). The lists of those peptide classifiers should be demonstrated in tables (or in supplement tables), in which they are arranged by their categories. These information may give some ideas to the readers even though they were not significant statistically.
<Minor points>
- In Figure 1, the font size of each graph is small and difficult to read. Please make it larger.
- In some references (ref, 5, 18, 31, 37, 51), the bibliographic information is missing
Responses
<Major points>
Response 1: The list of the 88 peptides discriminating active from inactive LN, as well as the list of the 177 peptides predictive for glomerular chronicity >=25%, are now provided in supplementary data for further research.
<Minor points>
Response 1 : We uploaded a new version of the figure 1 in higher resolution and making the front size of each graph larger (page 8).
Response 2: We completed the missing bibliographic informations from the references. Please note that we added some references in the manuscript and that the concerned references are now:
- ref 5 (Page 14 ; Line 420-421)
- ref 19 (Page 14 ; Line 448-449)
- ref 32 (Page 15 ; Line 482-484)
- ref 38 (Page 15 ; Line 495-496)
- ref 57 (Page 16 ; Line 545-546)
Reviewer 3 Report
Although the value of urinary peptidome to predict long-term CKD, or renal flares in SLE, remains to be evaluated, this reports' failure to improve on current practice is a valuable contribution to this special issue.
Author Response
Response to Reviewer 3 Comments
Dear reviewer,
Thank you for your precious time in reviewing our paper and for appreciating our work.
Sincerely,
Dr Maxence TAILLIAR & Pr Noémie JOURDE-CHICHE
Round 2
Reviewer 1 Report
The authors should discuss role of tubulointerstitial inflammation in lupus nephritis in the introduction. Pls use both human and animal studies (if needed ) to highlight the importance of tubular injury in lupus nephritis and progression to end stage renal disease.
Reviewer 2 Report
The authors revised the manuscript adequately according to the reviewer's suggestions. The supplementary tables are useful for the readers who are interested in this research field. I have no more suggestions for the authors.
